# Effects of Ozone Treatment on Health-Related Quality of Life and Toxicity Induced by Radiotherapy and Chemotherapy in Symptomatic Cancer Survivors

**DOI:** 10.3390/ijerph20021479

**Published:** 2023-01-13

**Authors:** Bernardino Clavo, Angeles Cánovas-Molina, Yolanda Ramallo-Fariña, Mario Federico, Delvys Rodríguez-Abreu, Saray Galván, Ivone Ribeiro, Susana C. Marques da Silva, Minerva Navarro, Damián González-Beltrán, Juan A. Díaz-Garrido, Sara Cazorla-Rivero, Francisco Rodríguez-Esparragón, Pedro Serrano-Aguilar

**Affiliations:** 1Research Unit, Hospital Universitario de Gran Canaria Dr. Negrín, 35019 Las Palmas de Gran Canaria, Spain; 2Chronic Pain Unit, Dr. Negrín University Hospital, 35019 Las Palmas de Gran Canaria, Spain; 3Radiation Oncology Department, Hospital Universitario Dr. Negrín, 35019 Las Palmas de Gran Canaria, Spain; 4Fundación Canaria Instituto de Investigación Sanitaria de Canarias (FIISC), 35019 Las Palmas de Gran Canaria, Spain; 5Universitary Institute for Research in Biomedicine and Health (iUIBS), Molecular and Translational Pharmacology Group, University of Las Palmas de Gran Canaria, 35016 Las Palmas de Gran Canaria, Spain; 6Instituto Universitario de Enfermedades Tropicales y Salud Pública de Canarias, Universidad de La Laguna, 38296 La Laguna, Spain; 7CIBER de Enfermedades Infecciosas (CIBERINFEC), Instituto de Salud Carlos III, 28029 Madrid, Spain; 8Spanish Group of Clinical Research in Radiation Oncology (GICOR), 28290 Madrid, Spain; 9Network for Research on Chronicity, Primary Care, and Health Promotion (RICAPPS), Instituto de Salud Carlos III, 28029 Madrid, Spain; 10Servicio de Evaluación y Planificación del Servicio Canario de Salud (SESCS), 38109 Santa Cruz de Tenerife, Spain; 11Instituto de Tecnologías Biomédicas (ITB), Universidad de la Laguna, 38296 La Laguna, Spain; 12Medical Oncology Department, Complejo Hospitalario Universitario Insular Materno-Infantil de Gran Canaria, 35016 Las Palmas de Gran Canaria, Spain; 13Medical Oncology Department, Hospital Universitario de Gran Canaria Dr. Negrín, 35019 Las Palmas de Gran Canaria, Spain; 14Psychiatry Department, Hospital Universitario de Gran Canaria Dr. Negrín, 35019 Las Palmas de Gran Canaria, Spain; 15Universidad de La Laguna, 38296 La Laguna, Spain

**Keywords:** ozone therapy, health-related quality of life, chemotherapy-induced side effects, radiation-induced side effects, cancer survivors, chemotherapy-induced neuropathy, anxiety and depression, pain

## Abstract

(1) Background: The continuous improvement in cancer treatment has led to improvement in patients’ survival and a subsequent increase in the number of cancer survivors living with adverse side effects of cancer treatments, sometimes with a high and adverse impact on their health-related quality of life (HRQOL). Side effects of cancer treatments are frequently associated with chronic status of oxidative stress, inflammation, and/or ischemia. The potential for ozone treatment to modulate those processes and improve some of those adverse effects has previously been described. The aim of this study was to evaluate the effect of ozone treatment on the HRQOL and grade of toxicity in symptomatic cancer survivors. (2) Methods: Before and after ozone treatment, we assessed (i) the HRQOL (according to the EQ-5D-5L questionnaire) and (ii) the grade of toxicity (according to the Common Terminology Criteria for Adverse Events of the National Cancer Institute of EEUU (CTCAE v.5.0)) in 26 cancer survivors with chronic side effects of radiotherapy and chemotherapy. (3) Results: There was a significant (*p* < 0.001) improvement in the EQ-5D-5L index as per the self-reported outcome evaluation of patients’ health status. All the dimensions of the EQ-5D-5L questionnaire (mobility, self-care, activities, pain/discomfort, and anxiety/depression) and the self-evaluation of the health status using the visual analog scale were significantly improved (*p* < 0.05). The grade of toxicity was also significantly decreased (*p* < 0.001). (4) Conclusions: In cancer survivors with chronic side effects of cancer treatment, ozone treatment can improve the grade of toxicity and the HRQOL. These results merit additional research. Further studies are ongoing.

## 1. Introduction

Diagnosis and treatment of cancer is continuously improving, offering better survival for these patients. However, as survival improves, the percentage of cancer survivors living with adverse effects of cancer treatments also increases. Moreover, some of these side effects become chronic and difficult to manage, especially for chronic pain, because of a scarcity of trials focusing on the long-term safety and effectiveness of interventions [1]. These uncontrolled side effects lead to a negative impact on physical and psychological performance, as well as on health-related quality of life (HRQOL). This is the case for highly incidental chemotherapy-induced peripheral neuropathy (CIPN), presented in 19–85% of cancer patients treated with potentially neurotoxic drugs [2], with a high prevalence after the end of chemotherapy of one-third of patients at six months or later [3], with limited therapeutical approaches of proven effectiveness [4,5,6]. Another example is radiation-induced chronic pelvic toxicity. Patients with uterine cervix or prostate cancer have an increased risk of radiation-induced symptoms because higher radiation doses are usually delivered in these tumors. Gastrointestinal symptoms after pelvic radiation can affect the quality of life in 50% of patients, and in 20–40%, the effect can be moderate or severe [7]. Chronic pelvic pain can affect 38% of long-term survivors and is associated with high overall mental and somatic morbidity [8]. Some authors have proposed pelvic-radiation-induced toxicity as a new disease, defined as “transient or long-term problems, ranging from mild to very severe, arising in non-cancer tissues resulting from radiotherapy treatment” [7]. For these reasons, the American Society of Clinical Oncology (ASCO) established a priority research area to develop novel strategies to mitigate the chronic side effects of cancer therapy [9].

Among the main processes involved in the production and perpetuation of chemo- or radiation-induced side effects are oxidative stress, inflammation, and ischemia (secondary to damage in micro-vessels and endarteritis) [10,11]. The main mechanisms of actions of ozone treatment (O_3_T) involve an adaptive response of the organism, leading to the improvement of (i) modulation of oxidative stress by enhancement of Nrf2 [12,13,14], (ii) a subsequent anti-inflammatory effect associated with pro-inflammatory cytokine modulation, and (iii) modulation of blood flow with potential decrease of ischemia/hypoxia [10,13,15]. These mechanisms could explain the successful treatment outcomes of O_3_T of one chronic radiation-induced lesion reported one century ago [16]. However, ozone is not a pharmaceutical drug; it is associated with a therapeutical procedure (health technology), randomized controlled trials (RCT) are difficult to carry out, and there have been almost no RCTs until the current century. We have previously published reports about the beneficial effects of adding ozone treatment to the management of difficult clinical conditions, such as painful CIPN [17], and several refractory radiation-induced pelvic symptoms, such as chronic pain [18,19], hematuria [20], and rectal bleeding [21,22]. Indeed, for the latter condition, O_3_T has been described as having a grade of recommendation 1C in the guidelines for the treatment of chronic radiation proctitis published by the American Society of Colon and Rectal Surgeons [23].

The aim of this report was to assess the impact of O_3_T on HRQOL and on toxicity in cancer survivors treated in a Chronic Pain Unit for persistent side effects of cancer treatment.

## 2. Methods

### 2.1. Patients and Study Design

Between June 2019 and June 2022, 54 patients with persistent or refractory side effects of cancer treatments were submitted to the Chronic Pain Unit for evaluation of compassionate ozone treatment. Eventually, 26 cancer survivors filled out the HRQOL assessment before and after O_3_T. Adjuvant O_3_T was administered in 15 patients affected by chemotherapy-induced neuropathy and in 11 patients with radiation-induced side effects.

All patients were provided with written information to request their consent to participate in the research project and to receive treatment, according to the Declaration of Helsinki of 1975. Both the research study and the individual compassionate ozone treatment used in these clinical conditions were evaluated by the Provincial Research Ethics Committee of Las Palmas, Spain (Ref. 2019-288-1). 

### 2.2. Ozone Treatment

Ozone (an O_3_/O_2_ mixture) was obtained from clinical-grade oxygen using two medical ozone generators (Ozonosan Alpha-plus^®^; Dr. Hänsler GmbH, Iffezheim, Germany, and Ozonobaric P, Sedecal, Madrid, Spain). According to the symptoms of the patients, the ozone treatment procedures were focused on (i) a systemic ozone effect, using rectal insufflation, with or without (ii) a local ozone effect, with ozone exposition of the damaged area (cutaneous wounds or intravesical). For ozone insufflations, the O_3_/O_2_ concentrations were progressively increased between 10 and 30 μg/mL, as previously described [17]. For topical administration, ozone concentrations usually ranged between 40 and 10 μg/mL, according to patients’ tolerance or on the basis of the presence or absence of local infection. Patients were treated with O_3_T if there was no evidence of tumor progression.

### 2.3. HRQOL Assessment

HRQOL was assessed with the Spanish version (v1.0, 2009) of the EQ-5D-5L questionnaire developed by the EuroQol Group [24,25]. The EQ-5D-5L is a preference-based measure of HRQOL that yields an index score anchored at 0 (dead) and 1 (full health) by assessing five different dimensions. Every dimension is measured according to the level of impairment/severity, ranging from 1 (“I have no problems”) to 5 (“I have a lot of problems”) in mobility, self-care, activities of daily living, pain or discomfort, and anxiety or depression. It also includes a visual analog scale (VAS) measuring self-perceived general health, ranging from “0” (worst health status) to “100” (best health status). 

### 2.4. Grade of Toxicity 

The grade of toxicity of cancer treatments was assessed according to the Common Terminology Criteria for Adverse Events (CTCAE v.5.0) of the National Cancer Institute of EEUU. Grade-0 is no toxicity; Grade-1 is toxicity without symptoms or mild symptoms; Grade-2 is moderate symptoms, limiting instrumental activity of daily living; Grade-3 is severe symptoms, limiting self-care activity of daily living, requiring hospitalization or elective medical intervention; Grade-4 is associated with life-threatening consequences, with urgent intervention indicated; and Grade-5 is death.

### 2.5. Statistical Analyses

The SPSS software package (version 15 for Windows, IBM, Armonk, NY, USA) was used for statistical analyses. All data were described as median (quartile 2) and quartiles 1 and 3 (Q1–Q3). The correlation between the grade of toxicity and EQ-5D-5L dimensions was assessed by Spearman’ rho. Paired comparisons (pre and post) were conducted with the exact (significance) Wilcoxon rank test. Comparisons for qualitative variables were conducted with the exact (significance) McNemar’s test. Though more conservative than asymptotic tests, exact tests were used due to the small sample size. *p*-values of <0.05 were considered statistically significant.

## 3. Results

The sex distribution of the patients included 11 males and 15 females, with a median age of 66 years (between 36 and 84 years). No significant differences were detected either by sex or age according to cancer treatment previously received, chemotherapy or radiotherapy (*p* = 0.246 and *p* = 0.750, respectively).

A total of 22 patients (85%) received systemic ozone treatment by rectal insufflation, 16 as an exclusive procedure, and 6 with additional local O_3_T. Ten patients (38%) received local O_3_T: three as an exclusive procedure and seven with additional systemic O_3_T, and eight by cutaneous application and two by intravesical application. One patient was treated by autohemotherapy. In three patients, O_3_T for the management of side effects was administered during the second or further line of chemotherapy or immunotherapy cancer treatment. The median duration of O_3_T was 17.5 weeks (Q1–Q3: 16–25). Overall, the median number of O_3_T sessions was 40 (Q1–Q3: 39–40).

Results on HRQOL and grade of toxicity are summarized in Table 1.

The HRQOL, analyzed by the EQ-5D-5L index, remained the same in five patients, and three of them already had the maximum score before O_3_T, despite symptoms with Grade-2 or Grade-3 toxicity (thus, there was no possible further improvement of the EQ-5D-5L index, regardless of the potential symptomatic improvement). On the contrary, the HRQOL worsened in three patients at the end of O_3_T. Thus, 18 patients (69%) improved their EQ-5D-5L index after O_3_T. 

The median EQ-5D-5L index was 0.67 (Q1–Q3: 0.31–0.92) at baseline and 0.85 (Q1–Q3: 0.63–1) at the end of O_3_T. This was a significant difference (*p* = 0.001). According to cancer treatment, for patients treated because of radiation-induced side effects, the EQ-5D-5L index was 0.82 (Q1–Q3: 0.32–0.92) at baseline and 0.92 (Q1–Q3: 0.62–1) at the end of O_3_T; this difference was not statistically significant (*p* = 0.139). For patients treated because of chemotherapy-induced side effects, the EQ-5D-5L index was 0.65 (Q1–Q3: 0.20–0.74) at baseline and 0.75 (Q1–Q3: 0.63–0.92) at the end of O_3_T; this was a significant difference (*p* = 0.002).

The domains that more frequently showed any impairment (levels 2 to 5) in our study group were “pain or discomfort” (in 69% of patients) and the “ability to carry out the activities of daily living” (in 65% of patients). All the domains showed improvement in > 60% of patients with any initial impairment, with the highest percentage of improvement in “pain or discomfort” (78%) and “anxiety or depression” (79%) of patients with any initial impairment. 

After O_3_T, changes including all patients (*n* = 26) were statistically significant for all dimensions of the EQ-5D-5L (see Table 1). The results of each dimension of the EQ-5D-5L in detail were as follows: (i) in mobility, 15 patients showed alterations before O_3_T, and 9 of them (60%) reported improvement after O_3_T; (ii) in self-care capacity, 12 patients showed alterations before O_3_T, and 8 (67%) reported improvement after O_3_T; (iii) in the ability to carry out the activities of daily living, 17 patients showed alterations before O_3_T, and 13 (76%) reported improvement after O_3_T; (iv) in pain or discomfort, 18 patients showed alterations before O_3_T, and 14 (78%) reported improvement after O_3_T; and (v) in anxiety or depression, 14 patients showed alterations before O_3_T, and 11 (79%) reported improvement after O_3_T (see details in Table 2).

In patients treated because of chemotherapy-induced side effects (*n* = 15), all EQ-5D-5L domains showed significant improvement after O_3_T: (i) mobility (*p* = 0.047), (ii) self-care capacity (*p* = 0.031), (iii) ability to carry out the activities of daily living (*p* = 0.004), (iv) pain or discomfort (*p* = 0.008), and (v) anxiety or depression (*p* = 0.016). In patients treated because of radiotherapy-induced side effects (*n* = 11), changes in the five EQ-5D-5L domains were not statistically significant.

Self-evaluation of the health status with the EQ VAS did not change in 4 patients, worsened in 1 patient, and improved in 21 patients (81% of patients). Overall, the median value was 57.5 (Q1–Q3: 40–73) before O_3_T and 77.5 (Q1–Q3: 65–90) at the end of O_3_T (*p* < 0.001). Health status was significantly improved in both groups of patients. In patients treated because of chemotherapy-induced side effects, the median value was 50 (Q1–Q3: 40–70) before O_3_T and 75 (Q1–Q3: 65–90) after O_3_T (*p* < 0.001). In patients treated because of radiation-induced side effects, the median value was 60 (Q1–Q3: 40–90) before O_3_T and 80 (Q1–Q3: 70–95) after O_3_T (*p* = 0.029) (Figure 1).

The results of the assessment of the grade of toxicity are summarized in Table 3.

Eighteen patients (69%) reported an improvement ≥50% of the treated symptoms (*p* < 0.001), three patients reported an improvement of 30%, and five patients reported no improvement of the treated symptoms.

The grade of toxicity according to the CTCAE v.5.0 classification decreased in 14 patients (54%). The median value was 2 (Q1–Q3: 2–3) before O_3_T and 2 (Q1–Q3: 1–2) after O_3_T (*p* < 0.001). In patients treated because of radiotherapy-induced side effects, the median value was 2 (Q1–Q3: 2–3) before O_3_T and 1 (Q1–Q3: 0–2) after O_3_T (*p* = 0.016). In patients treated because of chemotherapy-induced toxicity, the median value was 2 (Q1–Q3: 2–3) before O_3_T and 2 (Q1–Q3: 1–2) after O_3_T (*p* = 0.016) (Table 3, Figure 2). 

The grade of toxicity before O_3_T did not show a correlation with any dimension of the EQ-5D-5L. However, after O_3_T, a lower grade of toxicity was correlated with a higher ability to carry out the activities of daily living (rho 0.445, *p* = 0.023) and a higher improvement of the health status EQ VAS (rho -0.431, *p* = 0.028).

## 4. Discussion

This report showed that adjuvant treatment with O_3_T decreased the grade of toxicity and improved self-perceived HRQOL in cancer survivors with chronic or persistent symptoms secondary to cancer treatments.

There are very few studies assessing the effect of O_3_T in the management of side effects of cancer treatments, with most of them not focused on self-reported outcomes assessed by patients, such as QOL or symptoms directly related to QOL. One report of a RCT showed improvement of QOL (by the QLQ-C30 questionnaire) using O_3_T by major autohemotherapy and Viscum album therapy in patients with non-small cell lung cancer, but this effect was assessed during standard chemotherapy [26]. Another work using O_3_T by major autohemotherapy also described its beneficial effect in a group of cancer patients with fatigue who were treated with O_3_T during cancer treatment or in a palliative setting [27]. Thus, unlike in our study, those studies were not focused on QOL in cancer survivors with chronic/refractory side effects of cancer treatments.

Chronic pain is a frequent problem in cancer patients. In our study group, 69% of patients had pain or discomfort, 54% had anxiety or depression, and 77% suffered alterations in one or two of these HRQOL domains. These symptoms can be directly addressed by medical interventions, although sometimes conventional approaches are unsuccessful. Patients with advanced disease are usually managed in Palliative Care Units, and opioids can be used as required, in addition to other adjuvant analgesics. However, in cancer survivors without tumor progression, pain management can become difficult because (i) the use of opioids is controversial due to the side effects and lack of evidence and (ii) a neuropathic component frequently exists, which adds additional difficulties in pain management [28]. The improvement obtained in the “pain or discomfort” domain in this study agrees with our previous experiences with O_3_T in the management of refractory pelvic pain [18,19] and painful chemotherapy-induced peripheral neuropathy [17]. Currently, we are enrolling patients in a RCT for the latter symptom (NCT04299893), and a different RCT is planned for the former (EuraCT 2022-000320-37).

“Anxiety or depression” is another relevant domain for focused therapeutic approaches. From a psychological point of view, cancer involves a true “emotional tsunami” for the patient. Around 40% of cancer patients show some levels of clinical distress at some point in the disease process [29], and some combination of mood disorders occur in 30–40% of patients in hospital settings, without a significant difference between palliative-care and non-palliative-care settings [30]. These psychological alterations decrease the QOL and can impact the adherence to cancer treatment and the subsequent evolution of the patient [31,32]. Thus, it is relevant to research new tools for the detection and evaluation of psychological risk factors [33] and new therapeutic strategies. In this report, O_3_T also showed significant improvement in the anxiety or depression domain. It is probably the case that the main action of ozone treatment for this improvement could be related to the improvement of chronic symptoms. However, an additional direct action of systemic ozone treatment on anxiety and depression status cannot be dismissed. We have previously summarized how oxidative stress and increased proinflammatory cytokines can be involved with the pathogenesis of anxiety and depression by deregulation of brain functions and neuronal signaling [34]. On the contrary, some studies have described the fact that O_3_T can improve anxiety, depression, and sleep quality in non-cancer patients [35,36]. 

In our study, 73% of cancer survivors showed alterations in mobility, self-care capacity, or the ability to carry out the activities of daily living. We would like to highlight that these dimensions cannot be directly addressed by pharmacological interventions, and these three dimensions were also improved by O_3_T. It is possible that some degree of improvement in these dimensions could be dependent on the evolution of pain and symptoms after O_3_T. However, a RCT in patients with rheumatoid arthritis treated with methotrexate also showed that patients in the O_3_T group had an improvement in clinical scales associated with QOL as the “Disease Activity Score 28”and the “Health Assessment Questionnaire-Disability Index” [15,37].

Despite the fact that some authors reported a ceiling effect for EQ-5D-5L in patients with cancer undergoing curative [38] or palliative [39] treatment, our results confirm that EQ-5D-5L is sensitive to change in all dimensions, discriminating in the magnitude of the treatment response in most patients who, departing from worst baseline scores, improved considerably after the O_3_T administration. These results contribute to reinforcing those previously reported by authors who observed that patient-reported outcomes provided by generic HRQOL instruments, such as the EQ-5D, were as sensitive and responsive as specific scales for measuring patient-reported outcomes of functioning and well-being during follow-up after curative [40] and palliative [41,42] cancer treatments, complementing similar results in mortality, median survival, and use of healthcare resources [42].

Three patients had a worse QOL index after O_3_T. The grade of toxicity did not improve in any of these patients; however, the EQ VAS became better in one patient, remained the same in another patient, and got worse in one patient. Interestingly, the patient who got worse showed improvement of the initial symptoms until she underwent local tumor relapse, and new regional symptoms emerged that were related to tumor growth. It is necessary to take into account that the relevance of the symptoms and their transfer to the different EQ-5D-5L dimensions depends on the different self-perceptions of the patients and are not always exclusively associated with self-reported health status. This adds some difficulty to the assessment of patient-centered outcomes. It may be the case that EQ VAS could be one of the most relevant because it can summarize the overall perception of the patients about their different symptoms, medical problems, and personal repercussions. In our study, self-evaluation of the EQ VAS was improved in 81% of the patients, from a median value of 58 before to 78 after O_3_T. We believe that the probability and magnitude of potential improvement are clinically relevant and are in agreement with the comment by Stoker one century ago in *The Lancet*, when the results of the surgical uses of ozone were described as “…satisfactory from every standpoint, be it humanitarian, scientific, or economic” [43].

The main side effects of O_3_T in our study were (i) for rectal insufflation, middle and transient meteorism and bowel bloating (frequent) [21,35,44]; (ii) for local ozone application, local pruritus in a few patients, which disappeared after the O3 concentration was decreased; (iii) one patient treated by rectal insufflation mentioned a headache after O_3_T sessions 4 to 12, which was controlled with paracetamol. After the 12th session, the patient did not present cephalalgia anymore. Additional potential side effects and safety information have recently been published [44,45].

This study had several limitations. (i) This was a non-controlled study, and the placebo effect or natural symptom evolution could not be dismissed completely. However, most patients had chronic symptoms without clinical improvement for several months before the commencement of O_3_T. Another fact against the potential placebo effect was that the improvement in self-reported health status was significantly correlated with a significant decrease in the grade of toxicity. (ii) Another relevant limitation of the study was the limited sample size, preventing further detailed subgroup analysis, although it was enough to show significant differences in most of the analyzed variables, according to the subgroups of patients treated because of chemotherapy-induced or radiation-induced toxicity. Patients treated with O_3_T because of chemotherapy-induced side effects showed significant improvement in all domains of the EQ-5D-5L, as well as a significant decrease in the grade of toxicity. However, the smaller sample size of the subgroup of patients treated because of radiation-induced side effects (*n* = 11) could explain that improvements in the EQ-5D-5L domains were not statistically significant. Despite this, the improvement in health status EQ VAS and the decrease in grade of toxicity remained significant in this small group of patients. (iii) Patients were submitted to and treated with O_3_T because of different symptoms, especially in the radiation-induced group (hemorrhagic proctitis, hemorrhagic cystitis, chronic wounds, or local pain), and several tumor types (colon and rectum and gynecological tumors were the most frequent). 

We believe that the results of our study and the scarcity of alternative therapeutical approaches in most of our patients merit further research. In this way, as well as to overcome the current limitations, the assessment of HRQOL is one of the main variables in our ongoing (i) prospective study of O_3_T in patients submitted to our Chronic Pain Unit (NCT05417737), (ii) RCT in cancer survivors treated with O_3_T because of painful chemotherapy-induced peripheral neuropathy (NCT04299893), and (iii) RCT in cancer survivors treated with O_3_T because of chronic pelvic radiation-induced toxicity (EudraCT 2022-000320-37). 

## 5. Conclusions

This preliminary study evaluated cancer survivors treated with ozone treatment because of different persistent or refractory side effects of cancer treatments. Adjuvant treatment with ozone provided symptom improvement in a high percentage of patients, and it was associated with a significant improvement in the five evaluated dimensions of health-related quality of life that were self-reported by patients, alongside a significant decrease in toxicity grade. Further studies in this field are ongoing.

## Figures and Tables

**Figure 1 ijerph-20-01479-f001:**
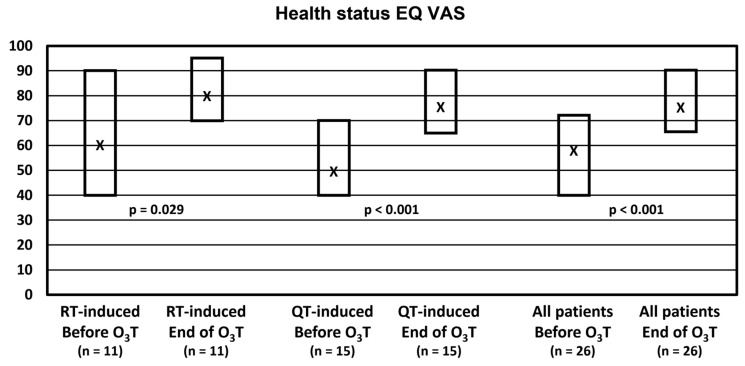
Self-reported health status in the visual analog scale (EQ VAS). X: median value. Box: quartiles 1 and 3.

**Figure 2 ijerph-20-01479-f002:**
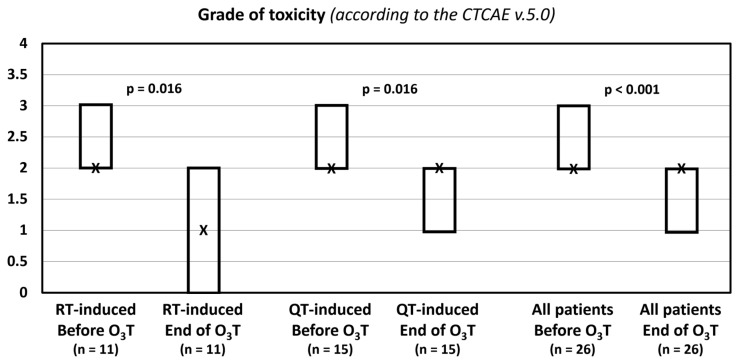
Grade of toxicity according to the Common Terminology Criteria for Adverse Events (CTCAE v.5.0) of the National Cancer Institute of EEUU. X: median value. Box: quartiles 1 and 3.

**Table 1 ijerph-20-01479-t001:** Health-related quality of life (HRQOL) and grade of toxicity. Q1–Q3: quartiles 1 and 3.

	Before-O_3_T Median (Q1–Q3)	After-O_3_T Median (Q1–Q3)	*p*-Value
EQ-5D-5L index	0.67 (0.31–0.92)	0.85 (0.63–1)	0.001
Mobility	2 (1–3)	1 (1–2)	0.028
Self-care capacity	1 (1–3)	1 (1–2)	0.008
Ability to carry out the activities of daily living	2.5 (1–3)	1 (1–3)	0.011
Pain or discomfort	3 (1–4)	1 (1–3)	0.002
Anxiety or depression	2 (1–3)	1 (1–2)	0.014
Health status EQ VAS	58 (40–73)	78 (65–90)	<0.001

**Table 2 ijerph-20-01479-t002:** Descriptive answer of each EQ-5D-5L dimension by time, *n* (%). MO: mobility; SC: self-care capacity; UA: ability to carry out the activities of daily living; PD: pain or discomfort; AD: anxiety or depression.

	pre-O_3_T					post-O_3_T				
	MO	SC	UA	PD	AD	MO	SC	UA	PD	AD
No problems	11 (42.3)	14 (53.8)	9 (34.6)	8 (30.8)	12 (46.2)	14 (53.8)	18 (69.2)	16 (61.5)	16 (61.5)	15 (57.7)
Slight problems	7 (26.9)	5 (19.2)	4 (15.4)	3 (11.5)	3 (11.5)	8 (30.8)	5 (19.2)	3 (11.5)	3 (11.5)	6 (23.1)
Moderate problems	4 (15.4)	4 (15.4)	8 (30.8)	5 (19.2)	7 (26.9)	3 (11.5)	3 (11.5)	4 (15.4)	4 (15.4)	4 (15.4)
Severe problems	2 (7.79	2 (7.7)	4 (15.4)	9 (34.6)	4 (15.4)	0	0	3 (11.5)	3 (11.5)	1 (3.8)
Unable/extreme problems	2 (7.7)	1 (3.8)	1 (3.8)	1 (3.8)	0	1 (3.8)	0	0	0	0

**Table 3 ijerph-20-01479-t003:** Grade of toxicity according to the Common Terminology Criteria for Adverse Events (CTCAE) v.5.0 of the National Cancer Institute of EEUU. * Q1–Q3: quartiles 1 and 3. ** Percentage of patients with improvement regardless of initially altered patients.

	Pre-O_3_T (Median, Q1–Q3) *	Post-O_3_T (Median, Q1–Q3)	*p*	Pre-O_3_T Patients Altered *n* (%)	Post-O_3_T ** Patients Improved *n* (%)
All patients	2 (2–3)	2 (1–2)	<0.001	26 (100%)	14 (54%)
RT-induced toxicity	2 (2–3)	1 (0–2)	0.016	11 (100%)	7 (64%)
QT-induced toxicity	2 (2–3)	2 (1–2)	0.016	15 (100%)	7 (47%)

## Data Availability

The raw data supporting the conclusions of this article will be made available by the authors under request.

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
