# Peer review of "Effects of Ozone Treatment on Health-Related Quality of Life and Toxicity Induced by Radiotherapy and Chemotherapy in Symptomatic Cancer Survivors"

_ijerph, 2023, doi:10.3390/ijerph20021479_

Round 1

Reviewer 1 Report

Very interesting topic related to a field where the advanced results in cancer survivors could create post treatment symptoms very difficult to manage. Ozone could represents a real help in reducing the post treatment side effects and increase the quality of life of these patients. The data are indicative for further positive results in the field and the Authors reported clearly all the observed response to the ozone treatment.

By a general point of view, to avoid conflicts with other medical specialties based on purely pharmacological or surgical treatments, I strongly suggest replacing the term therapy with treatment to emphasize the characteristics of an integrative and complementary approach with ozone which can be better defined as an oxidative medicine technique.

Please, clarify or explain this sentence (rows 118,119,120) regarding the local treatment concentrations. They injected subcutaneously, intramuscular or other? The concentration of 40 seems to be very high!

"For local administration, ozone concentrations usually ranged between 40 and 10 μg/mL according to patients’ tolerance or on the basis of the presence or absence of local infection."

Author Response

General comments

We deeply thank the reviewer for the positive comments about our study.

Specific comments:

Comment 1

From a general point of view, to avoid conflicts with other medical specialties based on purely pharmacological or surgical treatments, I strongly suggest replacing the term therapy with treatment to emphasize the characteristics of an integrative and complementary approach with ozone which can be better defined as an oxidative medicine technique.

  • We have replaced the term “therapy” with “treatment” throughout the text

Comment 2

Please, clarify or explain this sentence (rows 118,119,120) regarding the local treatment concentrations. They injected subcutaneously, intramuscular or other? The concentration of 40 seems to be very high!

  • We have not used subcutaneous or intramuscular ozone administration.

In lines 116-117, it was already stated “(ii) a local ozone effect, with ozone exposition of the damaged area (cutaneous wounds or intravesical)”.

However, to clarify this point, in line 119 of the revised version the term “local” was replaced by “topical” administration.

Reviewer 2 Report

Just consider removing the analysis of altered/not-altered patients as it introduces very confusing data. Please read my comments inside the text.

Author Response

We thank the reviewer for the comments and suggestions about our study which have allowed us to improve the manuscript.

In the revised version, we have taken into account the comments written by the reviewer inside the text:

  1. We have removed all comments about “altered patients” in Table 1 (in legends and columns) and we have rewritten the related comments in old lines 181-188 (lines 178-183 in the revised version)
  2. We have also removed the comparisons with Canary Islands population in Table 1 and lines 169-172, as well as the related reference (ref #26).
  3. According to the comment in lines 306-307, we have moved this information to line 168 (lines 165-168 in the revised version).
